# The Impact of College Students’ Academic Stress on Student Satisfaction from a Typological Perspective: A Latent Profile Analysis Based on Academic Self-Efficacy and Positive Coping Strategies for Stress

**DOI:** 10.3390/bs14040311

**Published:** 2024-04-11

**Authors:** Yibo Zhao

**Affiliations:** Institute of Education, Xiamen University, Xiamen 361005, China; 25720221152474@stu.xmu.edu.cn

**Keywords:** latent profile analysis, academic stress, student engagement, student satisfaction, academic self-efficacy, positive stress coping strategies

## Abstract

This investigation employs Latent Profile Analysis (LPA) to analyze data from 1298 Chinese university students, aiming to clarify the mechanisms through which individual psychological resources, primarily academic self-efficacy and positive coping strategies, affect student satisfaction in the context of academic stress. Four distinct profiles emerged based on levels of academic self-efficacy and positive coping strategies: Low-Spirited, General Copers, Capable but Passive, and Optimistic and Confident. These profiles demonstrate significant variances in the sources of academic stress, student engagement, and student satisfaction, with a ranking order from most to least satisfied as follows: Optimistic and Confident, Capable but Passive, General Copers, and Low-spirited. While academic stress uniformly augments engagement across all profiles, its effect on student satisfaction diverges—negatively for Low-spirited and General Copers, and positively for Capable but Passive and Optimistic and Confident. The analysis reveals varying levels of academic stress-tolerance among profiles, highlighting the critical role of academic self-efficacy and indicating a possible nonlinear relationship between student engagement and satisfaction. These findings enhance our comprehension of student satisfaction’s intricate dynamics and suggest strategies to alleviate academic stress and improve psychological well-being.

## 1. Introduction

Student satisfaction has consistently been a pivotal topic within higher education research, serving as a crucial metric for assessing the learning conditions of university students and the quality of higher education [1]. As early as the 1960s, scholars began to investigate students’ satisfaction levels with the quality of school education. In 1966, the introduction of the Cooperative Institutional Research Program (CIRP) scale by the University of California, Los Angeles, marked a pivotal advancement in measuring student satisfaction within American higher education, setting a precedent for later studies [2]. Subsequently, building on the American Customer Satisfaction Index (ACSI) model, Noel-Levitz devised the National Student Satisfaction Inventory (SSI) to systematically assess the satisfaction levels of university students across the United States, thereby integrating customer satisfaction principles into educational settings [3]. Martensen and colleagues have also employed the European Customer Satisfaction Index (ECSI) model for investigating student satisfaction [4]. Additionally, Chinese academia has also conducted in-depth research on the issue of student satisfaction and provided guidance recommendations for policy formulation. For example, in 2021, the Ministry of Education of China issued the Implementation Plan for the Review and Evaluation of Undergraduate Education in General Colleges and Universities (2021–2025), which clearly proposed that student satisfaction should be used as a key indicator to measure education quality assurance and teaching effectiveness and be evaluated [5]. In addition, the National Institute Of Education Sciences conducted a third round of national higher education satisfaction survey in 2021, and the survey results reflect the satisfaction status of college students in terms of learning experience and gains, professional identity, practical teaching opportunities, etc. [6]. Attaching importance to student satisfaction reflects the paradigm shift in higher education from focusing on academic value to focusing on student subjects and cultivation of talent [7], which is also a very important trend in China’s higher education after entering the massification stage until the current popularization stage.

Student satisfaction, as a subjective psychological experience, is influenced by various factors. On one hand, it is affected by student engagement [8] and motivation [9]; on the other hand, it is more directly impacted by sources of academic stress [10,11]. Academic stress can negatively affect university students’ development and decrease satisfaction, with academic self-efficacy and coping strategies moderating the impact of academic stress. Students with high levels of academic self-efficacy, who believe they can control academic stressors, tend to experience less negative impact from stress and consequently report higher satisfaction levels [12]. Furthermore, the adoption of positive coping strategies, such as problem solving and seeking social support, directly contributes to alleviating the negative repercussions of academic stress on student satisfaction [13,14]. This indicates that the mechanisms by which academic stress negatively affects students warrant further investigation. This study aims to categorize academic self-efficacy and positive strategies for coping with stress as individual psychological resources, employing them as clustering indices to examine how subgroups with varying psychological resources respond to academic stress and the ensuing effects on student satisfaction. This approach aims to provide targeted recommendations for alleviating academic stress and promoting psychological health development among university students.

### 1.1. Factors Influencing Student Satisfaction

Student satisfaction is a short-term attitude originating from the evaluation of a student’s educational experience [15]. Elliot and Shin define student satisfaction as the subjective evaluation by students of the outcomes and experiences of their education [16]. Additionally, it can be understood as a function of the relative level of experiences and perceived performance regarding educational service [17]. By considering all, student satisfaction represents a multidimensional concept, encompassing academic quality, social integration, administrative services, and personal development, all of which are subject to students’ subjective evaluation and cognition [18]. Marsh and colleagues have indicated that the primary variations in student satisfaction are predominantly found at the individual student level, rather than at the university or departmental levels, suggesting that these differences should be explained by other student-related variables [19]. Consequently, scholars have conducted extensive research to explore the relationship between student satisfaction and variables related to student learning. Aldemir and others proposed a framework for factors influencing satisfaction, encompassing four aspects: institutional factors, extracurricular activities, student expectations, and demographic factors [20]. Institutional factors include the quality of course instruction, atmosphere, resource facilities, and management services [21,22]. Extracurricular activities encompass students’ interpersonal interactions, club activities, and internships [23]. Student expectations refer to the students’ subjective desires regarding the quality of the institution’s operations and their educational experiences, with the discrepancy between expectations and experiences forming the basis of student satisfaction [24]. Demographic factors cover individual student characteristics such as gender, family background, age, and academic performance [25]. Cotton and colleagues introduced the “Happy-Productive Theory”, which suggests that student satisfaction is moderated by students’ anxiety and stress. Specifically, satisfaction increases when stress levels are low and decreases when stress levels are high [26]. Additionally, models have been developed to describe the relationship between student engagement and satisfaction. For instance, Kahu’s engagement model delineates a process where psychological and behavioral engagement acts as a mediator between the educational context and student outcomes, including satisfaction [27]. Similarly, Biggs’s 3P model (Presage, Process, Product) outlines how pre-existing conditions and students’ engagement with learning activities (Process) contribute to learning outcomes and satisfaction (Product) [28]. Building on Biggs’s 3P model, Guo and colleagues proposed a comprehensive model suggesting that students’ perceptions of the learning environment, influenced by personal characteristics and contextual factors, promote engagement in learning, which, in turn, affects academic outcomes, satisfaction, and generic skills [8]. According to these models and theories, engagement in learning and students’ perceptions of their environment are two key factors affecting student satisfaction.

### 1.2. Academic Stress, Student Engagement, and Student Satisfaction

The concept of student engagement is recognized as a process variable capable of predicting learning outcomes. It is influenced by antecedent variables (perceived environment, student background, and support) and, in turn, affects subsequent outcomes (grades, skills, and satisfaction) [23,27]. Other studies also propose that student engagement mediates the relationship between personal and environmental factors and learning outcomes [29,30]. Specifically, the environment plays a critical role in shaping learning outcomes, including student satisfaction. Students’ perceptions of the learning environment are considered key; as Asikainen and Entwistle noted, it is their perception of the learning environment, rather than the objective environment itself, that impacts their learning [30,31]. Students’ perceptions influence their learning approaches, engagement, academic achievement, and satisfaction. Academic stressors in the learning environment, exceeding students’ coping resources, present environmental demands and challenges that create burdens affecting academic tasks, peer relationships, and teacher–student interactions. Research indicates that academic stress not only triggers negative emotions, reducing happiness and satisfaction [32,33], but also leads to decreased attention and memory, increased absenteeism, and reduced engagement [34].

In college students, academic stress exerts the most direct influence on students, with this influence being significantly shaped by their cultural milieu. Li and colleagues found that Chinese university students primarily face academic pressures from low study efficiency and peer competition, while American students’ predominant academic stresses stem from examinations [35]. China’s educational system, with its emphasis on tests and rankings, creates a high-pressure environment [36], necessitating that students exert extra effort to outperform peers and secure superior positions. Such intense academic pressures can lead to overcommitment and academic involution, adversely affecting students’ physical and mental well-being [37].

Thus, the interaction between students and their environment shapes their experience (i.e., perception of academic stress), which then influences their learning behaviors and engagement, ultimately affecting satisfaction and learning outcomes [38]. Student engagement, as a process variable, is believed to mediate the relationship between sources of academic stress and student satisfaction.

### 1.3. The Impact of Academic Self-Efficacy and Positive Coping Strategies on Student Satisfaction

While previous research highlights the negative impact of academic stress on student learning, it has also spurred interest in exploring protective factors when students face academic stress, with individual psychological resources being a potential key aspect. These resources modulate the relationship between potential threats and stress responses, promoting better psychological adaptation and adjustment [39]. Two widely studied individual psychological resources are self-efficacy and coping strategies.

The cognitive appraisal theory of stress describes an individual’s perception of stressors as a two-stage cognitive evaluation process. Initially, in the primary appraisal phase, an individual assesses the potential impact of stressors, considering whether they pose a threat, challenge, or harm. This assessment is deeply personal, reflecting the individual’s subjective viewpoint. Subsequently, the secondary appraisal phase involves evaluating the resources they have available for coping with these stressors. This evaluation is crucial as it determines the individual’s perceived ability to manage or mitigate the stressor’s impact, complementing the initial appraisal by assessing coping capabilities and resources [40]. Self-efficacy plays a significant role in how individuals confront stress [41], exerting a distinct influence on the appraisal of stressors and the selection and implementation of coping strategies [42]. Characterized as a person’s belief in their ability to achieve desired goals or standards, self-efficacy modulates an individual’s cognition, motivation, and emotions [43]. In this sense, individuals with high self-efficacy tend to perceive potential stress situations as challenges rather than threats [12,41,44]. DeJonge and others argue that in academic contexts, it is more appropriate to measure domain-specific self-efficacy rather than a generalized form of self-efficacy [45]. Therefore, this study focuses on academic self-efficacy, considering it as one of the individual psychological resources.

However, research by Jex et al. indicates that self-efficacy serves as a partial moderator between stressors and stress responses [46], suggesting that while self-efficacy can reduce the negative impact of stressors, its effectiveness is contingent upon the presence of other factors, such as coping strategies [47]. Specifically, stress-coping and self-efficacy are related, rather than being independent psychological resources. On one hand, individuals with high self-efficacy perceive stressors as challenges and believe in their capability to overcome difficulties. Yet, without the employment of positive coping strategies, the effectiveness of self-efficacy may not be fully realized [47]. On the other hand, individuals possessing positive and effective coping strategies, but lacking the belief in their ability to overcome the impact of stressors, may perceive these strategies as challenges, leading to an avoidance of difficulties [48]. Further research, utilizing latent profile analysis, has identified four distinct subgroups among university students: students with a low generalized use of coping strategies, students with a predominance of social coping strategies, students with a predominance of cognitive coping strategies, and students with a high generalized use of coping strategies. This study found that students with high-stress coping strategies scored the highest in general self-efficacy [39]. Therefore, it can be inferred that students who proactively and positively cope with academic stress are likely to have higher levels of self-efficacy.

### 1.4. The Present Study

In summary, this study constructs a structural equation model with sources of academic stress as environmental variables, and student engagement and satisfaction as behavioral and outcome variables, respectively. Employing a person-centered approach, academic self-efficacy and positive stress coping strategies are used as clustering indices in Latent Profile Analysis (LPA) to identify the interactive effects of these variables and differentiate types of student subgroups. The study ultimately aims to explore the mechanisms of satisfaction formation among students with different psychological resource subgroups when confronted with academic stress. To clarify the analysis further, this study proposes the following research hypotheses:

**H1:** 
*There are distinct profiles based on variations in academic self-efficacy and positive stress coping strategies.*


**H2:** 
*Different profiles of students exhibit variations in sources of academic stress, student engagement, and satisfaction.*


**H3:** *The mechanisms of satisfaction formation differ among profiles*.

## 2. Materials and Methods

### 2.1. Participants and Procedure

On the basis of ensuring the scientific design of the survey and considering principles of feasibility and cost-effectiveness, this study employs a convenience sampling method. This approach is efficient, straightforward, easily accessible, and cost-effective. The study’s participants comprised 1298 Chinese students from freshman to senior years, with an average age of 22.56 years (rang 18.51–26.54 years, SD = 1.32). Participants completed the survey via an online platform (https://www.wjx.cn, accessed on 15 November 2023) containing the research inventory. To protect privacy and ensure data anonymity, the questionnaire design avoided questions that could directly identify individuals, such as names, addresses, or other personal information. The online survey system processed submissions anonymously, thereby eliminating the possibility of direct traceability. Participants may independently complete this survey at their convenience. However, they are only able to submit their responses after answering all questions, and they are provided with sufficient time (typically about 15 min) to complete the inventory.

In terms of demographics, the sample consisted of 39.4% male and 60.6% female students. Freshmen accounted for 42.1% of the participants, sophomores 27.0%, juniors 23.6%, and seniors or above 7.3%. Regarding academic disciplines, 49.5% of the participants were from Humanities/Social Sciences (including economics, management, law, education, literature, history, philosophy, arts, and military science), and 50.5% from Science/Engineering fields (including natural sciences, engineering, agriculture, and medicine). All participants provided informed consent prior to their involvement and were informed of the study’s purpose. Participation was voluntary, with participants free to withdraw from the survey at any time.

### 2.2. Measures

#### 2.2.1. Academic Stress

Referencing the academic stress section of the College Stress Scale developed by Solberg et al. [49], items were selected and adapted to reflect the actual conditions of Chinese university students. The survey investigated the frequency of academic stress from sources such as examination pressures and coursework burdens through four questions. For instance, items included statements like, “Competing academically with classmates causes me significant stress” and “Under the competitive pressure from classmates, I invest energy in my studies that far exceeds the required or expected amount”. All items on the scale used a Likert five-point scoring system, with higher scores indicating greater levels of stress experienced by the students.

#### 2.2.2. Positive Stress Coping

Select the positive stress coping part of the stress coping scale developed by Carver et al. [50], the survey employed seven items to explore how university students actively seek solutions when faced with stressors. For example, questions included statements such as, “When I encounter stress or problems, I tend to approach them positively” and “When faced with stress or problems, I actively strive to find solutions”. All items on this scale were measured using a Likert five-point scoring system, where higher scores indicate a more proactive approach by students in coping with stress.

#### 2.2.3. Academic Self-Efficacy

The academic self-efficacy portion of the Motivated Strategies for Learning Questionnaire (MSLQ) developed by Pintrich et al. was used to investigate university students’ confidence in their academic studies through three items [51]. For instance, one item states, “Regardless of my academic performance, I never doubt my learning capability”. Items were scored on a Likert scale of five points, with higher scores indicating stronger academic self-efficacy among students.

#### 2.2.4. Student Engagement

Used the College Student Learning Engagement Scale developed by Guo [23], comprising six dimensions. These dimensions include course study, student-faculty interaction, peer interaction, extracurricular activity, and deep learning approach, all of which fall under the category of behavioral learning engagement. Course study was assessed through three items, investigating students’ involvement in classroom learning processes, such as “Paying close attention and actively thinking during class”. Student–faculty interaction included three items, examining the degree of students’ academic interactions with teachers, like “Proactively communicating with teachers after class”. Peer interaction encompassed three items, reflecting students’ engagement in collaboration and communication with classmates, for instance, “Working with other students to complete coursework or projects”. Extracurricular activities were explored through five items, assessing students’ engagement in learning activities outside of the classroom, such as “Participating in scientific research projects or competitions”. Deep learning approach was assessed with six items, focusing on students’ pursuit of a deeper level of learning, like “Connecting current learning with past experiences. “Additionally, the scale addressed affective learning engagement with three items on college student school belonging, probing students’ psychological identification and emotional investment in their institution, exemplified by “I feel that I am a part of my school”.

#### 2.2.5. Student Satisfaction

Used College Student Satisfaction Scale developed by Guo [23], which includes seven items to assess student satisfaction during their university tenure. This survey explored students’ satisfaction with classmates, teachers, courses, administration, and overall experience. The scale used a Likert five-point scoring system, with higher scores indicating more positive evaluations by students in aspects such as course organization, teaching quality, peer interaction, campus cultural environment, and learning management and services.

### 2.3. Data Analysis

This study used SPSS 27.0 for data entry, organization, and preliminary analysis, and Mplus 8.3 for Latent Profile Analysis, mediation effects, and multi-group analysis. The analytical approach was divided into four steps: First, confirmatory factor analysis was conducted on the survey data to test the reliability and validity of the measurement scales. Second, descriptive statistics were computed, and the correlations among various variables were analyzed. Third, using academic self-efficacy and positive stress coping strategies as manifest variables, a latent profile model was established. The optimal category model was determined based on model fit indices such as the Akaike Information Criterion (AIC), Bayesian Information Criterion (BIC), and Entropy. Differences in engagement, its dimensions, and satisfaction across profiles were compared using the BCH method. Fourth, the influence mechanisms and mediation effects on student satisfaction among different profiles of university students were analyzed using structural equation modeling and bias-corrected Bootstrap tests.

Additionally, before conducting the formal data analysis, the Harman single-factor test was used to check for common method bias. A single common factor was set, and confirmatory factor analysis was performed using Mplus. The results indicated poor model fit, suggesting that the data in this study did not have a severe common method bias issue.

### 2.4. Ethics

Human subjects were used in this study, so we followed the Helsinki Declaration and its subsequent amendments. This study has been approved by the Ethical Committee of Xiamen University, Xiamen, China. Informed consent was obtained from participants before the data collection. All participants were free to withdraw from the study at any time and the confidentiality of their responses was assured. Upon completion of the questionnaire by each participant, a small token of appreciation was provided as a gesture of thanks for their efforts.

## 3. Results

### 3.1. Confirmatory Factor Analysis

To ensure the reliability and validity of the research instruments, an initial confirmatory factor analysis (CFA) was conducted with a model comprising ten primary factors. The results indicated a good fit for the measurement model. Moreover, factor loadings for each item were all above 0.60, with z-values significant at the 0.001 level. The Average Variance Extracted (AVE) values for all factors exceeded 0.50, and the square roots of the AVE values were greater than the inter-factor correlation coefficients, demonstrating satisfactory discriminant validity. Additionally, all factors exhibited Cronbach’s alpha coefficients and composite reliabilities exceeding 0.70, indicating good reliability. Overall, the survey scales used in this study exhibit a desirable level of reliability and validity, making them suitable for more in-depth analysis (see Table 1).

### 3.2. Descriptive Statistics and Correlation Analysis

The means, standard deviations, and correlation coefficients for academic stress sources, student engagement, student satisfaction, positive coping strategies, and academic self-efficacy are presented in Table 1. The average score for academic stress among students was slightly above the theoretical midpoint of 3 (M = 3.19), indicating that university students experience a moderate level of stress originating from parents, teachers, peers, and academic examinations. However, students generally have confidence in their academic abilities (M = 4.13) and tend to adopt positive strategies for coping with stress (M = 3.70). The highest score in student engagement was observed in the dimension of peer interaction (M = 4.37). Students also scored relatively high in course learning (M = 4.33), teacher–student interaction (M = 3.93), extracurricular activity (M = 3.20), deep learning approach (M = 4.09), and university belonging (M = 4.23). These scores suggest that students who are highly engaged in classroom activities and teacher–student communication are inclined towards a deeper understanding and exploration of their studies, and possess a strong emotional identification with their institution. The overall student satisfaction was favorable (M = 4.07), indicating that, in general, university students have a positive subjective experience during their time at school, with their personal needs being adequately met.

Correlation analysis revealed significant relationships between all variables. The correlations between sources of academic stress and various dimensions of student engagement, student satisfaction, and academic self-efficacy were relatively weak. Moderate positive correlations were found between different dimensions of student engagement and student satisfaction, positive coping strategies, and academic self-efficacy.

### 3.3. Determining the Number of Latent Profiles

Latent Profile Analysis (LPA) was conducted using scores from academic self-efficacy and positive coping strategies for stress as variables. In these models, the two variables were designated as uncorrelated within each profile, and variances across profiles were assumed to be equal to prevent convergence issues. The most appropriate number of profiles was determined using the following criteria: (1) Competing models were evaluated using the Akaike Information Criterion (AIC), Bayesian Information Criterion (BIC), and the sample size-adjusted Bayesian information criterion (aBIC), with smaller values indicating improved model fit [52]. (2) The classification accuracy within the LPA models was assessed by estimating entropy (ranging from 0 to 1), with an entropy value above 0.8 indicating that the classification accuracy reached 90% [52,53]. (3) Significant Lo–Mendell–Rubin Likelihood Ratio Test (LMR) and Bootstrap Likelihood Ratio Test (BLRT) results suggested a significant difference between the k-class model and the k − 1 class model. In addition to fitting statistical criteria, it was also essential to ensure that the profiles had theoretical and substantive significance. Therefore, it was decided that each profile should consist of at least 5% of the total sample size to be retained for analysis [54].

Positive coping strategies and academic self-efficacy were inputted as indicators into the Latent Profile Analysis (LPA) to determine the optimal number of individual characteristic profiles. Table 2 presents the fit indices for the LPA model: Log-Likelihood (LL), Akaike Information Criterion (AIC), Bayesian Information Criterion (BIC), and sample size-adjusted Bayesian Information Criterion (aBIC) values decreased with an increasing number of profiles. Additionally, the entropy values of each profile model were greater than 0.8, indicating that the classification accuracy was above 90%. When the model reached four and five profiles, the values of LL, AIC, BIC, and aBIC were lower, and the results of the Lo–Mendell–Rubin (LMR) and Bootstrap Likelihood Ratio Test (BLRT) supported the optimal fit for both four and five profiles. Consequently, models with two and three profiles were no longer considered. However, in the five-profile model, the smallest group had an insufficient sample size (<5%), and its interpretability and theoretical significance were less robust than the four-profile model. Therefore, considering the balance between model simplicity and accuracy, the four-profile model was ultimately selected as the best fit.

The results of the Latent Profile Analysis (LPA) model were analyzed to characterize and label different profiles. Figure 1 illustrates the average scores for two indicators across three profiles. Based on the scoring characteristics of the four profiles in the two personal psychological resource indicators, these profiles were named as follows: Low-Spirited, General Copers, Capable but Passive, and Optimistic and Confident. Students in the Low-Spirited category exhibited lower scores in both positive coping strategies and academic self-efficacy, comprising 34.30% of the sample. General Copers displayed average scores in both dimensions, accounting for 17.50% of the sample. The Capable but Passive group showed lower scores in positive coping strategies but higher in academic self-efficacy, representing 10.40% of the sample. Lastly, Optimistic and Confident students scored high in both indicators and formed the largest group, at 37.80% of the sample.

### 3.4. Testing for Differences in Outcome Variables across Different Profiles

After determining the optimal model, the study used the refined Bolck–Croom–Hagenaars (BCH) method to examine the performance of each profile on the outcome variables and conducted pairwise comparisons [55]. Table 3 presents the overall chi-square test results for pairwise comparisons between profiles and the BCH results adjusted for chi-square values. According to the overall chi-square test, significant differences existed between the four profiles (*p*s. < 0.001).

Specifically, students in the Low-Spirited group showed the lowest performance in terms of student engagement across various dimensions, student satisfaction, and sources of academic stress compared to the other profiles. Except for academic stress sources and extracurricular activity where there were no significant differences with the Capable but Passive group, their performance was significantly lower in all other aspects compared to the remaining profiles (*p*s. < 0.001). General Copers scored significantly lower than both the Capable but Passive group and the Optimistic and Confident group across all variables (*p*s. < 0.001). Additionally, the Capable but Passive scored significantly lower than the Optimistic and Confident type in all variables (*p*s. < 0.001).

### 3.5. Multi-Group Analysis of Student Satisfaction across Different Profiles

A multi-group structural equation modeling approach was used to explore the varying mechanisms that affect student satisfaction across different profiles (see in Table 4). Initially, in the direct relationship between sources of academic stress, student engagement, and student satisfaction, the academic stress sources significantly positively predicted student engagement in both the Low-Spirited and General Copers groups (β = 0.551, *p* < 0.001; β = 0.548, *p* < 0.001). However, student engagement significantly negatively predicted student satisfaction in these profiles (β = −0.625, *p* < 0.001; β = −0.152, *p* < 0.001), with a significant difference existing between the two profiles in the “student engagement—student satisfaction” pathway (β = −0.473, *p* < 0.001,95%CI: −0.597–−0.377). For Capable but Passive and Optimistic and Confident subgroups, academic stress sources also significantly positively predicted student engagement (β = 0.305, *p* < 0.001; β = 0.241, *p* < 0.001), and student engagement significantly positively predicted student satisfaction (β = 0.358, *p* < 0.001; β = 0.386, *p* < 0.001). No significant differences were found between these two subgroups in these direct pathways (β = 0.064, *p* = 0.350, 95%CI: −0.039–0.211; β = −0.029, *p* = 0.796, 95%CI: −0.192–0.229). For all profiles, the influence of academic stress sources on student satisfaction was not significant (*p*s. > 0.05).

Secondly, regarding indirect effects, student engagement significantly mediated the relationship between academic stress sources and student satisfaction across all four profiles of students. Specifically, in the Low-Spirited and General Coper groups, student engagement negatively mediated the relationship between academic stress sources and student satisfaction (β = −0.344, *p* < 0.001; β = −0.083, *p* = 0.050), with a significant difference in mediation effects between the two profiles (β = −0.261, *p* < 0.001, 95%CI: −0.316–−0.209). In Capable but Passive and Optimistic and Confident, student engagement positively mediated this relationship (β = 0.109, *p* < 0.001; β = 0.093, *p* < 0.001), with no significant difference in mediation effects between these groups (β = 0.016, *p* = 0.446, 95%CI: −0.024–0.060).

The significance of these indirect effects was tested using a bias-corrected Bootstrap method. In this study, data were resampled 2000 times. The 95% confidence intervals (CI) for all mediation effects did not include zero, indicating that the indirect effects were significant (Table 4).

## 4. Discussion

Employing a multifaceted research approach, this study leveraged Latent Profile Analysis to categorize students based on their psychological resources, conducted tests of differences to highlight variations in student satisfaction across these profiles, and applied multi-group structural equation modeling to intricately examine how different types of psychological resources interact with academic stress to influence student engagement and satisfaction. This comprehensive methodological framework allowed for a deeper exploration into the complex processes that underlie the formation of student satisfaction in the context of varying psychological resources.

Firstly, based on the levels of academic self-efficacy and positive stress coping strategies, university students can be categorized into four profiles: Low-Spirited, General Copers, Capable but Passive, and Optimistic and Confident. Low-Spirited students exhibit low self-efficacy, believing they lack the ability to complete academic tasks, leading to a lack of motivation to proactively address academic stress. General Copers demonstrate moderate levels of both positive stress coping and academic self-efficacy. They have some confidence in their ability to complete academic tasks but maintain a degree of self-doubt, leading them to attempt to resolve and cope with academic stress. Capable but Passive students, despite their high self-efficacy, may experience a discrepancy between their self-perceived capabilities and their actual stress management strategies. This discrepancy can lead to procrastination or avoidance behaviors, potentially stemming from a fear of failure or perfectionist tendencies that paradoxically inhibit action [56]. Optimistic and Confident students are confident in their ability to complete academic tasks and actively respond to stress, taking measures to overcome it. In this study, Optimistic and Confident students emerged as the most prevalent group, indicating a positive trend towards proactive stress management among participants. However, it is important to note that such trends may vary in different educational contexts and cultural backgrounds. Chinese college students demonstrate distinctive approaches to managing academic stress relative to their international peers [35]. Notably, they prefer proactive and adaptive strategies, including confronting stressors directly and seeking support, while seldom opting for assistance-seeking or self-blame [57]. This tendency may stem from the heightened independence and autonomy characteristic of the current generation of Chinese students, who have their own perspectives and suitable strategies for the developmental challenges encountered. Further studies are warranted to explore the variance in academic stress coping strategies among diverse university student cohorts.

Secondly, the four student categories differ significantly in their scores on sources of academic stress, various dimensions of student engagement, and overall satisfaction. In descending order, the ranking is Optimistic and Confident, Capable but Passive, General Copers, and Low-Spirited. The Optimistic and Confident group, despite experiencing the highest levels of academic stress, demonstrates superior engagement and satisfaction, indicating a robust mechanism for coping with stress that aligns with Lazarus’s stress appraisal and coping theory [40]. Lazarus and Folkman defined coping as the cognitive and behavioral efforts used to manage stressors perceived to surpass an individual’s personal capabilities. This theoretical framework clarifies the process by which individuals first perceive and subsequently internally assess stress, ultimately deciding on their coping response after evaluating the coping resources at their disposal. The Optimistic and Confident students’ adeptness at employing constructive coping mechanisms, such as engaging with peers and instructors, not only helps them manage stress effectively but also enhances their academic engagement and satisfaction. These strategies appear to transform their perception of stress from a negative force to an opportunity for growth and learning. Conversely, Low-Spirited students, who feel a diminished sense of control over stressors, tend to lean towards less effective coping strategies, experiencing lower satisfaction as a result [58]. This contrast underscores the pivotal role of coping strategies and the perception of control in navigating academic stress and achieving student satisfaction.

Thirdly, in the nuanced dynamics of academic stress, student engagement, and satisfaction, the roles of academic self-efficacy and coping strategies emerge as pivotal. Engagement acts as a mediator in this relationship for all profiles. Interestingly, engagement increases across the board in response to academic stress. Yet, its impact on satisfaction diverges, enhancing satisfaction for Optimistic and Confident and Capable but Passive students, but diminishing it for the Low-Spirited and General Copers, hinting at a nuanced threshold in stress response. The variance in academic self-efficacy between these profiles—lower for Low-Spirited students and General Copers and higher in the Capable but Passive and Optimistic and Confident groups—is telling. High self-efficacy, especially when paired with constructive coping strategies that do not include avoidance, appears crucial in softening stress’s blow [46,59]. This interplay aligns with Hobfoll’s Conservation of Resources theory, which posits that individuals with ample personal resources like positive stress coping can initiate a virtuous cycle of resource acquisition and preservation, thereby insulating themselves against stress [60]. For Low-Spirited and General Copers, engagement may stem more from a compulsory sense of duty rather than a proactive choice, possibly due to their lower confidence in task completion or tasks surpassing their skills, leading to decreased satisfaction. This scenario underscores the potential nonlinearity of the engagement-satisfaction relationship, suggesting that it may be influenced by both the learning environment and the students’ personal resource arsenal. Such findings underscore the importance of considering both the intensity and nature of engagement activities alongside student capacities for stress management and resilience in fostering optimal educational outcomes.

## 5. Educational Implications

The increasing recognition of academic stress as a significant psychosocial issue among college students underscores its widespread prevalence and potential adverse effects. This study’s pivotal contribution lies in identifying positive stress coping strategies and academic self-efficacy as key psychological resources in mitigating academic stress. It has been found that students with higher levels of academic self-efficacy may have a higher threshold for facing academic stress, leading to greater efficiency in their study engagement and higher levels of student satisfaction. To mitigate academic stress’s negative impacts and bolster students’ learning quality, satisfaction, and well-being, the study recommends the following strategies. Firstly, “Low-Spirited” students often struggle with motivation due to low self-efficacy and therefore need to rebuild their self-perception to enhance confidence in their abilities. Incrementally setting achievable goals and engaging in small-scale academic projects or group activities can build confidence and self-efficacy, encouraging proactive academic stress management. Teachers should offer positive reinforcement and support, tailoring goals to ensure students’ success, while university administrations should emphasize mental health education and support services to foster an inclusive learning environment.

Secondly, “General Copers” exhibit a positive yet sometimes doubtful approach to stress. Enhancing self-confidence through constructive self-dialogue, setting realistic study plans, and ensuring learning continuity are crucial. Teachers play a vital role in providing strategic learning and coping advice, encouraging exploration of new methods, and maintaining a supportive yet challenging environment. Universities should facilitate collaborative study groups, offer extensive learning resources, and maintain a diverse evaluation system to support these students.

Thirdly, “Capable but Passive” students may exhibit high self-efficacy but engage in procrastination or avoidance under stress. Recognizing the detrimental effects of such behaviors, students should set clear objectives and practice effective time management. Teachers should stimulate interest with engaging content and foster participation and motivation through personalized feedback and challenging tasks. Universities could enhance engagement through career counseling, extracurricular activities, and practical project opportunities.

Lastly, “Optimistic and Confident” students, characterized by a robust belief in their academic abilities and an active coping approach to stress, should continue to pursue new challenges and maintain a balanced lifestyle. Teachers can offer advanced learning opportunities, guiding deep reflection and critical thinking, while universities should provide talent development programs and research opportunities to broaden these students’ horizons.

Implementing tailored strategies for different student types can significantly enhance their ability to manage academic stress, promoting academic and personal growth.

## 6. Conclusions Limitations and Future Studies

This study elucidates the complex mechanisms shaping university student satisfaction, revealing that satisfaction not only mirrors the extent of engagement in prior learning activities but also acts as a gauge for educational quality. Elevated satisfaction levels signal a deeper engagement in learning, suggesting that higher levels of student satisfaction could denote enhanced educational experiences. Yet, the equation of satisfaction extends beyond engagement to encompass the academic stress encountered and the arsenal of individual psychological resources at students’ disposal. Academic stress prompts an uptick in learning engagement as students tackle scholastic challenges, though such engagement isn’t always a product of volition. In instances where academic pressures surpass students’ coping capacities, engagement may escalate out of necessity rather than choice, potentially diminishing satisfaction. Herein lies the significance of positive stress coping strategies and academic self-efficacy, with students endowed with these qualities displaying a more robust tolerance for academic stress. Those with elevated academic self-efficacy, in particular, are prone to perceive stress as an opportunity for self-affirmation and achievement, thereby bolstering engagement and, by extension, satisfaction. Despite the overall high satisfaction reported, the study underscores the need for educators to discerningly address the varied responses of different subgroups to academic demands, advocating for the adoption of customized teaching strategies.

This study acknowledges its limitations, primarily stemming from its reliance on cross-sectional survey data, which, while illuminative of variable interrelations, falls short of establishing causality. The dynamic nature of the variables in question calls for longitudinal research to delve deeper into these interactions over time. Furthermore, the investigation into how academic stress impacts student satisfaction needs expansion, particularly in defining the stress thresholds that students can endure before experiencing adverse effects on engagement and satisfaction. Future research is encouraged to explore these thresholds and the nuanced interplay between stress, engagement, and satisfaction, especially under conditions of excessive stress. Additionally, future studies also should prioritize conducting in-depth interviews and focus group discussions to uncover the nuanced perspectives and personal experiences of students. This qualitative approach will enable researchers to capture the richness and complexity of students’ feelings, thoughts, and attitudes towards their academic environment, revealing insights that are not readily accessible through quantitative measures alone.

## Figures and Tables

**Figure 1 behavsci-14-00311-f001:**
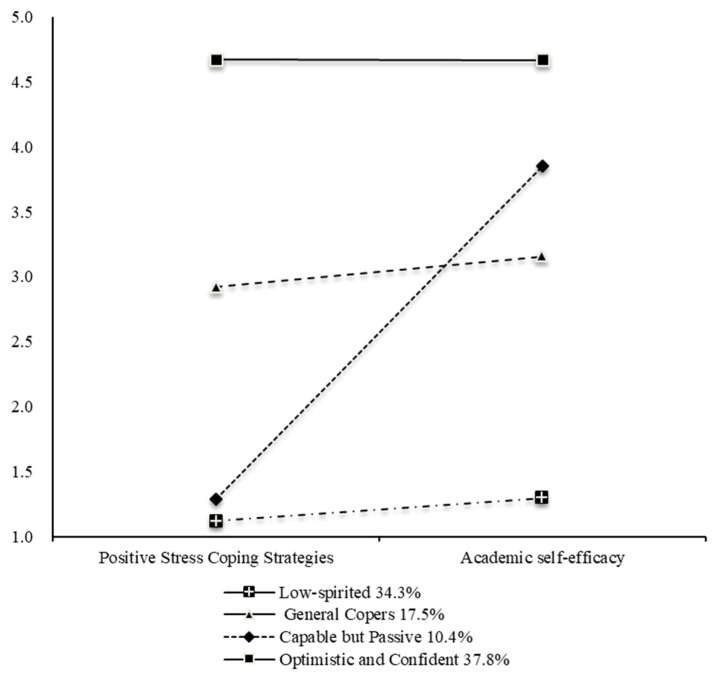
LPA Classification Results.

**Table 1 behavsci-14-00311-t001:** Correlation, Reliability, and Descriptive Analysis.

Variables	1	2	3	4	5	6	7	8	9	10	Cronbach’s α	CR	AVE	CFA Loadings Range
Academic stress	0.84										0.9	0.91	0.71	0.78–0.87
Course study	0.07 *	0.77									0.79	0.81	0.59	0.60–0.89
Student-faculty interaction	0.11 ***	0.64 ***	0.93								0.95	0.95	0.86	0.91–0.94
Peer interaction	0.08 **	0.58 ***	0.54 ***	0.89							0.92	0.92	0.8	0.88–0.90
Extracurricular activity	0.21 ***	0.41 ***	0.67 ***	0.35 ***	0.88						0.94	0.95	0.77	0.81–0.92
Deep learning approach	0.07 *	0.68 ***	0.74 ***	0.61 ***	0.62 ***	0.89					0.96	0.96	0.8	0.88–0.91
University belonging	0.06 *	0.67 ***	0.65 ***	0.63 ***	0.51 ***	0.84 ***	0.86				0.89	0.9	0.74	0.83–0.98
Student satisfaction	0.09 **	0.59 ***	0.58 ***	0.69 ***	0.43 ***	0.67 ***	0.70 ***	0.89			0.96	0.96	0.79	0.86–0.93
Positive Stress Coping Strategies	0.59 ***	0.27 ***	0.28 ***	0.29 ***	0.18 ***	0.32 ***	0.34 ***	0.38 ***	0.81		0.93	0.93	0.65	0.71–0.90
Academic self-efficacy	0.06 *	0.60 ***	0.67 ***	0.54 ***	0.53 ***	0.73 ***	0.70 ***	0.62 ***	0.34 ***	0.87	0.9	0.91	0.76	0.81–0.92
Mean	3.19	4.33	3.93	4.37	3.20	4.09	4.23	4.07	3.7	4.13				
SD	1.06	0.75	1.02	0.78	1.37	0.83	0.79	0.83	0.95	0.81				

Note: The lower triangular matrix represents the Pearson correlation coefficients among variables, while the diagonal values denote the square root of the Average Variance Extracted (AVE). CR = composite reliability. * *p* < 0.05, ** *p* < 0.01, *** *p* < 0.001.

**Table 2 behavsci-14-00311-t002:** Fit Indices for the Latent Profile Analysis (LPA) Model of Different Profiles.

K	LL	AIC	BIC	aBIC	Entropy	pLMR	BLRT	Class Probability
1	−4902.69	9813.39	9834.06	9821.35				1
2	−4071.12	8156.25	8192.43	8170.19	0.866	<0.001	<0.001	0.464/0.536
3	−3640.42	7300.84	7352.52	7320.76	0.935	<0.001	<0.001	0.264/0.347/0.389
4	−3364.98	6755.97	6823.16	6718.86	0.949	<0.001	<0.001	0.104/0.175/0.343/0.378
5	−3031.96	6095.92	6178.62	6127.79	0.989	<0.001	<0.001	0.039/0.170/0.237/0.254/0.300

Note: LL = loglikelihood; AIC = Akaike Information Criterion; BIC = Bayesian Information Criterion; aBIC = Sample-size Adjusted Bayesian Information Criterion; pLMR = *p*-value associated with the adjusted Lo–Mendel–Rubin likelihood ratio test; BLRT = Bootstrap Likelihood Ratio Test.

**Table 3 behavsci-14-00311-t003:** The relationships between the four profiles and outcome variables.

Variables	Profile1	Profile2	Profile3	Profile4	BCH χ^2^	Profile1	Profile1	Profile1	Profile2	Profile2	Profile3
	(n = 115)	(n = 362)	(n = 281)	(n = 540)		vs.	vs.	vs.	vs.	vs.	vs.
						Profile2	Profile3	Profile4	Profile3	Profile4	Profile4
Academic stress	3.57 (0.09)	3.70 (0.05)	4.45 (0.04)	4.87 (0.03)	628.73 ***	139.76 ***	1.42	468.28 ***	78.33 ***	64.66 ***	195.57 ***
Course study	3.36 (0.09)	3.67 (0.04)	4.42 (0.05)	4.90 (0.02)	906.17 ***	136.36 ***	8.33 ***	640.61 ***	103.67 ***	85.82 ***	204.99 ***
Student-faculty interaction	3.29 (0.10)	3.58 (0.05)	4.43 (0.05)	4.91 (0.02)	932.34 ***	165.21 ***	6.17 *	692.12 ***	100.41 ***	80.05 ***	237.61 ***
Peer interaction	3.19 (0.10)	3.59 (0.05)	4.42 (0.05)	4.94 (0.02)	985.08 ***	141.84 ***	11.22 ***	657.34 ***	115.65 ***	93.25 ***	278.61 ***
Extracurricular activity	3.50 (0.09)	3.68 (0.05)	4.50 (0.04)	4.93 (0.02)	890.89 ***	161.76 ***	2.81	629.13 ***	94.50 ***	79.49 ***	230.80 ***
Deep learning approach	3.34 (0.10)	3.67 (0.05)	4.45 (0.05)	4.90 (0.02)	836.83 ***	142.70 ***	8.22 **	602.78 ***	99.57 ***	71.38 ***	230.46 ***
University belonging	3.37 (0.10)	3.64 (0.05)	4.48 (0.05)	4.93 (0.02)	933.13 ***	154.48 ***	5.44 *	656.14 ***	100.74 ***	73.16 ***	239.84 ***
Student engagement	3.02 (0.13)	3.53 (0.05)	4.43 (0.05)	4.79 (0.03)	570.99 ***	145.22 ***	13.27 ***	401.57 ***	108.22 ***	35.12 ***	189.16 ***
Student satisfaction	2.94 (0.12)	3.47 (0.05)	4.45 (0.05)	4.90 (0.02)	980.43 ***	194.15 ***	17.12 ***	682.41 ***	145.79 ***	68.14 ***	281.12 ***

Note: profile1 = Low-spirited, profile2 = General Copers, profile3 = Capable but Passive, profile4 = Optimistic and Confident; * *p* < 0.05, ** *p* < 0.01, *** *p* < 0.001; The relationships between the four profiles and various outcome variables are represented by M(SE).

**Table 4 behavsci-14-00311-t004:** Path Coefficients Across Different Profiles.

Profiles	Path	Point Estimate	Product of Coefficients			BOOTSTRAP 2000 Times 95%CI	
			S.E.	Z	*p*	Lower	Upper
Low-spirit	Academic stress → student engagement	0.551	0.036	15.205	<0.001	0.501	0.638
	Student engagement → student satisfaction	−0.625	0.035	−18.06	<0.001	−0.692	−0.578
	Academic stress → student engagement → student satisfaction	−0.344	0.033	−10.52	<0.001	−0.441	−0.323
General Copers	Academic stress → student engagement	0.548	0.056	9.774	<0.001	0.465	0.655
	Student engagement → student satisfaction	−0.152	0.065	−2.324	0.020	−0.296	−0.095
	Academic stress → student engagement → student satisfaction	−0.083	0.042	−1.976	0.050	−0.194	−0.052
Capable but Passive	Academic stress → student engagement	0.305	0.045	6.79	<0.001	0.262	0.404
	Student engagement → student satisfaction	0.358	0.059	6.101	<0.001	0.237	0.439
	Academic stress → student engagement → student satisfaction	0.109	0.015	7.462	<0.001	0.092	0.129
Optimistic and Confident	Academic stress → student engagement	0.241	0.06	4.042	<0.001	0.173	0.322
	Student engagement → student satisfaction	0.386	0.11	3.52	<0.001	0.125	0.523
	Academic stress → student engagement → student satisfaction	0.093	0.025	3.732	<0.001	0.04	0.124

## Data Availability

The data presented in this study are available on request from the author.

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
