# Peer review of "The Impact of College Students’ Academic Stress on Student Satisfaction from a Typological Perspective: A Latent Profile Analysis Based on Academic Self-Efficacy and Positive Coping Strategies for Stress"

_behavsci, 2024, doi:10.3390/bs14040311_

Round 1

Reviewer 1 Report

Comments and Suggestions for Authors

Thank for the opportunity to review the manuscript entitled, "The Impact of College Students' Academic Stress on Student Satisfaction from a Typological Perspective: A Latent Profile Analysis Based on Academic Self-Efficacy and Positive Stress Coping Strategies." The specific methods are solid and very well reported. I very much appreciate the clarity and completeness of the communication of fairly complex results. Although the challenging of reviewing latent profile analyses is that there are many data analytical decisions that the authors need to make. It is nearly impossible to articulate the justification for all of the decisions. In general, I believe that the author provided a reasonable rational for factors and additional analyses based on latent factors. 

Special notice is that I have never used entropy metrics to any great degree in this form of modeling. But the author does a nice job of setting the stage for this metric to make sense and add to the data presented. That is not easy to do. 

The bigger issue is whether this study contributes to to theory or knowledge. Although this is an excellent review of previous literature on the topic it is not clear what theories are being tests or developed. Without a model or theory, then this is a collection of data from which latent factors are derived. The full meaning of these factors is unclear. To say that this information will lead to interventions is a bit misleading. Only the longest and most tenuous line of inference can lead one to take the data and results presented here and use it to develop any form of intervention. 

And every instance of the work "utilize" should be "use."  

My bias is that this form of modeling with explicit ties to theory are not that useful. Yet, these papers are widely published and this paper is clear and well put together. Also there some high quality presentation of results that others need to emulate. 

Author Response

Dear Reviewer,

We are submitting the revised version of our manuscript, ID: behavsci-2913947, titled "The Impact of College Students' Academic Stress on Student Satisfaction from a Typological Perspective:A Latent Profile Analysis Based on Academic Self-Efficacy and Positive Stress Coping Strategies" for further consideration in Behavioral Sciences. We greatly appreciate the constructive feedback provided by the reviewers and yourself, which has been instrumental in enhancing the quality and clarity of our work.

In response to the comments received, we have meticulously revised our manuscript to address the concerns and suggestions. Below, we briefly summarize the major revisions made:

  1. Clarification and Expansion of Theoretical Framework: We have elaborated on the theoretical underpinnings of our study, specifically addressing the application and development of stress appraisal and coping theory within the context of Chinese context. This aims to clarify how our research contributes to existing knowledge and theory in the field.
  2. Methodological Details: We have provided additional details regarding our research methodology, including more comprehensive information on participant selection, data collection procedures, and analysis techniques to enhance the reproducibility and transparency of our study.
  3. Data Analysis and Interpretation: We revised our analysis section to incorporate a deeper, more nuanced interpretation of our findings. This includes the consideration of alternative explanations and the potential implications of our results for both theory and practice.
  4. Response to Specific Comments: For each comment provided by the reviewers, we have included a point-by-point response detailing how we addressed the issue within the manuscript. These responses are attached as a separate document for your convenience.

We are grateful for the opportunity to revise our work and re-submit it for your consideration. Please find attached the revised manuscript along with a detailed response to the reviewers' comments. We hope that our revisions meet the approval of the reviewers and look forward to the possibility of our work being published in Behavioral Sciences.

Thank you very much for your time and consideration.

Reviewer 2 Report

Comments and Suggestions for Authors

-In your literature review, consider adding more relevant sources related to the Chinese context in relation to student satisfaction. What other studies in the Chinese context might be beneficial to add to the literature review? Academic stress may also vary depending on geographic context.   

-I would expand a little more on what student satisfaction entails in the introduction paragraph. Is it merely just student engagement and motivation as well as the impact of academic stress? 

-In the methodology, you provided the average age, but it may be helpful to include the range of ages also. 

-Be specific about how the survey was administered. Was it during class time, or could participants complete it independently? 

-Sampling needs to be discussed more specifically. 

-You mentioned in the first paragraph in the materials and methods section that participants ranged from freshman to senior years. In the next paragraph, you stated that there were seniors "or above." Did this study only include undergraduate students or were there also graduate students?

-Why were the participants only from two predominant fields? You may want to specify the selection criteria more specifically.

 -For the measurement scale, you mentioned the scale was selected and adapted to reflect the "actual conditions of Chinese university students." I think more contextual and background information on Chinese students specifically is needed in the literature review, especially since there is a lot of research on academic stressors and competition in the Chinese education system.  

-Were the scales developed by Guo designed for the Chinese higher education context? If so, you may want to state this since context is very important in this study. 

-Overall, the discussion is written clearly and concisely. However, it would be helpful to have more connections to other studies within the Chinese higher education context. 

-I would consider more practical implications in your conclusion. 

-Future research could also include qualitative data such as interviews or focus groups to gain more insights into student satisfaction within this specific context. 

-Check your references carefully. There are some spacing issues, especially with authors' initials. I am not an expert on this referencing style, but there appear to be some inconsistencies in the references.  

Author Response

(The authors gave the same response as above.)
